# Biohybrid Soft Robots Powered by Myocyte: Current Progress and Future Perspectives

**DOI:** 10.3390/mi14081643

**Published:** 2023-08-20

**Authors:** Zheng Yuan, Qinghao Guo, Delu Jin, Peifan Zhang, Wenguang Yang

**Affiliations:** 1School of Electromechanical and Automotive Engineering, Yantai University, Yantai 264005, China; yuanzheng0213@163.com (Z.Y.); g1260394153@163.com (Q.G.); 2School of Human Ities and Social Science, Xi’an Jiaotong University, Xi’an 710049, China; delujin@126.com; 3Control Science and Engineering, Naval Aviation University, Yantai 264001, China

**Keywords:** biological design, manufacturing techniques, myocyte-driven robots

## Abstract

Myocyte-driven robots, a type of biological actuator that combines myocytes with abiotic systems, have gained significant attention due to their high energy efficiency, sensitivity, biocompatibility, and self-healing capabilities. These robots have a unique advantage in simulating the structure and function of human tissues and organs. This review covers the research progress in this field, detailing the benefits of myocyte-driven robots over traditional methods, the materials used in their fabrication (including myocytes and extracellular materials), and their properties and manufacturing techniques. Additionally, the review explores various control methods, robot structures, and motion types. Lastly, the potential applications and key challenges faced by myocyte-driven robots are discussed and summarized.

## 1. Introduction

Robotics have continued to develop rapidly in recent decades, with various types of robots appearing one after another. The practicality and efficiency of robots have made them indispensable in various fields [1,2,3,4]. The component structures of robots are actuators, control systems, and sensing devices [5]. While the application areas of robots have increased dramatically, there are also more stringent requirements for the functions and performance of robots. Intelligent microrobots have been created for delicate tasks that are difficult to perform with traditional robots. They can simulate the shape and movement of real creatures with fast and efficient movements to perform operations that are difficult for traditional robots to perform.

Microrobots can be defined as mobile devices with a length of micrometers to centimeters [6,7,8]. Based on the existing driving mode of micro-bionic robots, they can be divided according to the traditional rigid robots [9,10,11], flexible material-driven robots [12,13,14], and biomaterial-driven robots [15,16,17,18]. Traditional rigid robots are made of hard materials that are characterized by high output power, high speed, high accuracy, and easy manipulation. However, traditional robots are complex, less flexible, and less agile [19,20]. Moreover, the mechanical work conversion efficiency of conventional electromechanical systems is low (<30%) and leads to significant heat loss [21]. Miniature conventional rigid robots have defects such as poor reliability, a relatively short service life, and low energy efficiency, and thus cannot meet some specific human needs to a certain extent. New drive methods using flexible functional materials have largely solved the shortcomings of traditional rigid robots. Flexible material-driven robots are primarily composed of synthetic materials with flexible properties. Common flexible material actuators include dielectric elastomer actuators (DEAs) [22,23,24,25], shape memory polymers (SMPs) [26,27,28], and liquid crystal elastomers (LCEs) [29,30,31,32]. Flexible material-driven robots can be driven by external stimulus control at scales of millimeters [33,34,35]. Their light mass, high adaptability to target shapes, low contact collision forces with the environment, and self-healing capabilities allow them to largely avoid tissue damage when interacting with biological tissues [36,37,38]. However, due to the lack of in-depth research on their stress, response speed, efficiency, and lifetime, research on flexible material actuators is still in its infancy, with difficulties regarding accurate modeling, low power density, and high drive voltage, which still hinder their practical applications [39].

**Figure 1 micromachines-14-01643-f001:**
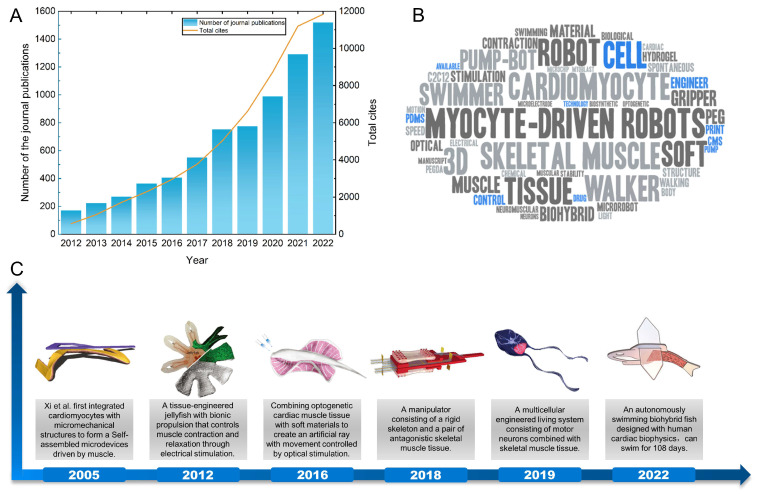
The development of myocyte-driven robots in recent years. (**A**) Number of journal publications and citations of research on biohybrid robots in recent years. (**B**) Key word cloud map on myocyte-driven robots. (**C**) Examples of typical achievements in the development of myocyte-driven robot research. Self-assembled microdevices driven by muscle. Reproduced from Reference [40] with permission from Nature Materials. A tissue-engineered jellyfish with biomimetic propulsion. Reproduced from Reference [41] with permission from Nature Biotechnology. Phototactic guidance of a tissue-engineered soft-robotic ray. Reproduced from Reference [42] with permission from Science. Biohybrid robot powered by an antagonistic pair of skeletal muscle tissues. Reproduced from Reference [43] with permission from Science Robotics. Neuromuscular actuation of biohybrid motile bots. Reproduced from Reference [44] with permission from Proc Natl Acad Sci USA. An autonomously swimming biohybrid fish designed with human cardiac biophysics. Reproduced from Reference [45] with permission from Science.

With the development and combination of disciplines such as bioengineering and 3D bioprinting, the combination of living biological actuators and nonliving biomaterials has become a possible solution to overcome the limitations of existing actuation methods [46,47,48,49]. In recent years, they have become a hot topic of research, as shown in Figure 1A. Biomaterials consist of biological materials or cellular communities embedded in a self-regenerating matrix of their own or artificial scaffolds. Similar to natural materials such as bone and muscle, biomaterials have the functional properties of living organisms and can grow, self-assemble, and self-heal when needed [50]. Compared with traditional rigid-actuated and flexible material-actuated robots, biohybrid robots can better simulate the microstructure and motion patterns of real living organisms. While biomaterials may have flexible properties, the difference between a flexible material-driven robot and a biomaterial-driven robot is the origin of the material and the underlying design principles. Flexible material-driven robots are designed based on mechanical principles, while biomaterial-driven robots are designed based on biological principles [51]. The driving sources of biomaterial-driven robots can be classified as myocytes [52,53], dorsal vascular tissue (DV tissue) from insects [54,55], or micro-organisms, among others (sperm, T cells, etc.) [56,57,58]. Among the various biohybrid robots, myocyte-driven robots are more mature in terms of research and manufacturing technology. They have remarkable controllability, output force, and power density, as well as self-assembly, self-healing ability, and better biocompatibility [43,59,60,61]. As long as the muscle tissues are given appropriate nutrients (e.g., glucose, adenosine triphosphate) in the culture environment, these muscle tissues can convert chemical energy into mechanical energy at a high energy conversion rate (≥50%) to provide energy for the robot [21,62]. While myocyte-driven robots offer exciting possibilities for integrating biological and mechanical systems, they also present significant challenges in terms of control, maintenance, durability, and ethical considerations. The field is still in its early stages, and ongoing research is addressing these and other questions. We compared the advantages and disadvantages of the three robots, as shown in Table 1. The various advantages are an important direction for future research (Figure 1B,C).

In this article, we provide a systematic review of myocyte-driven robots from different perspectives, as shown in Figure 2. First, we introduce two types of myocytes as a source of power for myocyte-driven robots, summarizing the cellular force, size, and controllability of cardiac and skeletal myocytes. Second, we discuss the extracellular materials used for myocyte-driven robots. The properties and fabrication methods of extracellular materials determine not only the performance of the myocytes (differentiation, contractility, and survivability) but also the performance of myocyte-driven robots (speed, force, and manipulation). Third, we review the control methods applied to myocyte-driven robots, including electrical stimulation, optical stimulation, and chemical stimulation, and the advantages and disadvantages of each of them. Then, the functions and applications of myocyte-driven robots are summarized according to their different modes of locomotion, including swimmers [41,44,63], walkers [64,65,66], grippers [52,67,68], and pump-bots [69,70,71], and we describe their performance and development process. Finally, we discuss and summarize potential applications and future challenges for research into myocyte-driven robotics.

## 2. Myocytes

Currently, the primary myocytes utilized in myocyte-driven robots are cardiomyocytes and skeletal myocytes. Both cardiac and skeletal muscles belong to the transverse muscle category and share similar contraction mechanisms. These muscles contain myogenic fibers composed of thick and thin myofilaments, which are aligned parallel to the cell’s long axis. As the cytoplasmic calcium ion concentration increases, calcium ions bind to troponin, initiating the binding and sliding of transverse bridges on the thick myofilaments toward the thin myofilaments. This process ultimately leads to the contraction of cardiac and skeletal myocytes [74,75]. In this chapter, we will describe the properties and acquisition methods of each of the two types of myocytes. The types of motion, materials, properties, and control methods for myocyte-driven robots are summarized in Table 2.

### 2.1. Cardiomyocytes

Cardiomyocytes have two major properties, namely, electrophysiological properties and mechanical properties. Sequential activation and inactivation of inward (Na^+^ and Ca^2+^) and outward flowing K^+^-carrying ion channels in the cardiomyocytes’ cell membrane allows cardiomyocytes to generate action potentials [87]. Microelectrode arrays (MEAs) are biosensors that can be used to monitor the electrophysiological activity of cardiomyocytes. When cardiomyocytes spontaneously generate action potentials, transient transmembrane potentials and ionic currents are generated, which polarize the electrodes by reconstructing the distribution of charge at the electrode–electrolyte–cell interface, causing a change in the electrode’s potential [88]. Desbiolles et al. recorded the action potentials within rat cardiomyocytes monolayers with a nanopatterned volcano-like microelectrode, demonstrating the autorhythmicity of cardiomyocytes (Figure 3A) [89]. The generation of electrical excitation by cardiomyocytes leads to cardiac contraction, a process known as excitation–contraction coupling (ECC), which achieves the conversion of chemical signals into mechanical energy, a specific electromechanical integration property possessed by cardiomyocytes [90]. Ca^2+^ plays a crucial role in ECC, coupling ES and mechanical contraction by regulating Ca^2+^ movement inside and outside the cardiomyocytes [91]. Cardiomyocytes have diameters of 10 ~ 25 μm and lengths of up to 100 μm [92]. By edge detection of cardiomyocytes, it is known that the contraction amplitude of cardiomyocytes is 2–15 μm, and the contraction amplitude of cardiomyocytes is significantly influenced by the extracellular Ca^2+^ concentration (Figure 3B) [93,94]. Microscopic observations of the beating process of free stem-cell-derived cardiomyocytes allowed an assessment of the detailed deformation of individual cardiomyocytes, and a beating frequency of up to 66.34 bpm was observed [95].

The contractile motion of cardiomyocytes correspondingly generates a certain contractile force. Oyunbaatar et al. tested the contractile force of cardiomyocytes by a polydimethylsiloxane (PDMS) column array with a surface pattern, which allowed the cardiomyocytes to grow directionally over microgrooves. The test results showed that the contractile force of the cardiomyocytes arranged on the microgrooves was 20% higher than that of the cardiomyocytes without microgrooves, indicating that the application of geometric stimulation can enhance the contractile force of cardiomyocytes (Figure 3C) [96]. Kim et al. proposed a silicone rubber cantilever device integrated with a high-sensitivity PDMS-encapsulated crackle sensor to orient cardiomyocytes on the cantilever’s surface grooves, and the contractile force of the cardiomyocytes was measured to be approximately 107 nN (Figure 3D) [97]. Because cardiomyocytes have the ability to conduct excitation, the cardiomyocytes can grow to the point of mutual contact to the extent that the electrical gap junctions cause the cardiomyocytes to contract synchronously [21,98]. Yin et al. measured the contractile force of individual cardiomyocytes based on moving magnetic beads to 10 μN [99]. By adhering cardiomyocytes to a gelatin hydrogel with microgrooves to form cardiac tissue, the cardiac tissue can generate a contractile stress of 10–50 kPa [100].

### 2.2. Skeletal Muscles

Skeletal muscles are the primary actuator in many organisms, and their inherent modularity and scalability make them a natural part of many cellular systems. Skeletal muscle consists of muscle fibers, each of which in turn consists of thousands of myofibrils and contains billions of myofilaments. When myofilaments are grouped together in a very ordered pattern, a sarcomere is formed, which is the basic contractile unit of skeletal muscle [75]. Myogenic cell precursors fuse to form multinucleated contractile myotubes. In vivo, skeletal muscles are controlled by motor neurons, and when motor intent is required, this motor intent is transmitted by the nervous system to the effector muscles via signals from the brain [101]; then, the motor neurons release acetylcholine (ACh) upon the activation of action potential. ACh causes the depolarization of the sarcolemma and the opening of the calcium channels in the sarcolemma and sarcoplasmic reticulum. The opening of the ion channels allows the calcium ions to enter the myotubes, activating the actin–myosin contractile machinery, which switches the myotubes from a resting (closed) to a contracted (open) state [102,103]. Myotubes can generate much higher contractile forces than cardiac myocytes [104]. Using the FEM-SPH coupled modeling technique, Vannozzi et al. found that skeletal muscle myotubes can generate a contraction force of 0.4 μN [105]. Akiyama et al. developed an engineered electrical stimulation culture system of skeletal muscles consisting of a gel culture mold, a medium replacement unit, and an electrical stimulation unit. The skeletal muscle tissue was measured by a force transducer and was found to produce a contraction force of more than 2.5 mN with a single pulse of electrical stimulation at 1 Hz (Figure 4A) [106]. Moreover, skeletal myocytes have the ability to regenerate after injury, whereas cardiomyocyte tissue does not [107]. Thus, skeletal muscle tissue has a broader range of adaptability and controllability.

The differentiation of myoblasts (C2C12) is a critical step in the manufacture of skeletal myocyte-driven robots. Typically, C2C12 cells can be differentiated into contractile myotubes using induced horse serum differentiation [108,109]. To achieve the goal of increasing the degree of differentiation of C2C12 cells and, thus, the contractility of skeletal muscle tissue, different methods have been developed to induce cell alignment. These include the use of groove/ridge micro-/nanopatterned substrates [110,111,112], electrical stimulation [113,114], optical stimulation and optogenetics [115,116,117], and chemical stimulation [118]. For example, Asano et al. applied optical stimulation sequences to C2C12 myotubes by genetically engineering the expression of the channel retinoid stromal variant, the channel retinoid green receptor (ChRGR). After training the myotubes through optical stimulation, muscle proteins were routinely aligned at regular intervals in contrast to untrained myotubes. The myotubes exhibited a distinct striation pattern and a significant increase in the number of contractile myotubes after training by optical stimulation (Figure 4B) [115]. The alignment of myogenic cells can be promoted by the external stimulation described above, which improves the myotubular differentiation rate and cellular contractility and has a direct correlation with the performance of skeletal myocyte-driven robots.

In recent years, neuromuscular systems that combine skeletal myocytes with motor neurons have emerged as novel biological drivers [44]. Cultivating engineered skeletal muscle fibers also with motor neurons leads to the formation of acetylcholine receptors at the interface of the two cell types. Because the skeletal muscle fibers in the neuromuscular system have structures similar to those of living skeletal muscle, such as independent structure and orientation, it can make a wide range of contractions in a single direction in response to neurotransmitters released by activated neurons (Figure 4C) [119]. Such biological actuators fused from two or more cell types may enable intelligent sensing and intelligent control functions and become a future direction for myocyte-driven robots.

**Figure 4 micromachines-14-01643-f004:**
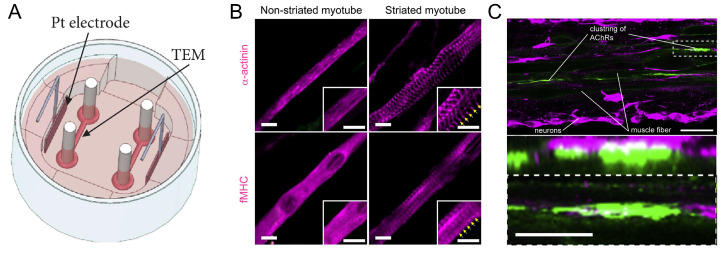
Properties of skeletal muscles. (**A**) Engineered skeletal muscle electrical stimulation culture system. Reproduced from Reference [106] with permission from Cyborg and Bionic Systems. (**B**) Effects of optical stimulation training on sarcomere assembly in ChRGR-expressing C2C12 myotubes. Reproduced from Reference [115] with permission from Sci Rep. (**C**) Formation of acetylcholine receptors at the point of contact between skeletal muscle fibers and neurons. Reproduced from Reference [119] with permission from Biomaterials.

## 3. Extracellular Materials for Myocyte-Driven Robots

In addition to muscle tissue, another important component of myocyte-driven robots is extracellular materials [112,120,121]. Extracellular materials provide structural support, a growth environment, and attachment substrates for myocytes. Since myocytes need to be attached or embedded in extracellular materials to build adaptive and biomimetic hybrid robots, extracellular materials must meet specific requirements, including excellent biocompatibility, desirable mechanical properties, and a tunable microstructure. This also has a direct impact on the myocyte’s state and robot performance [77,122,123,124]. Non-biological materials used for myocyte-driven robots are usually classified into three types: bioinert polymers [124,125], hydrogels (bioactive hydrogels [126,127,128,129] and artificial hydrogels [130]), and tissue-harvesting biomaterials [59,131], as shown in Figure 5. The properties of several extracellular materials differ, and we compare them in Table 3.

### 3.1. Bioinert Polymers

Bioinert polymers are chemically and physically inert in biological environments and exhibit exceptional stability and mechanical properties. PDMS is an ideal silicone elastomer with excellent resistance to biodegradation, along with biocompatibility, chemical stability, and permeability [132,133,134]. It also has good mechanical properties and is easy to handle and manipulate. Since myocytes need to be attached to PDMS, die-casting, surface coating, film cutting, and 3D bioprinting are often required to fabricate their structures to make them biologically active [77,86,135,136]. Holley et al. developed a self-stabilizing swimming robot with a fin propulsion mechanism. The robot used a composite PDMS material for the base and a thin cantilever for more precise control, with density modulation achieved by adding microballoons or nickel powder. The robot exhibited a unique propulsion pattern based on the angle of repose of its “fins” or cantilevers. The robot’s dive depth, pitch, and roll could be maintained without external intervention. Its maximum speed reached 142 μm/s [77]. Tanaka et al. poured a suspension of neonatal rat cardiomyocytes into PDMS micromolds and cultured them for 7 days. The cardiomyocytes formed several contractile bridges on the sidewalls of the microchannel. The smallest ever ultramicrofluidic oscillator for pumping was made (Figure 6A) [86]. Hasebe et al. used a biomixing actuator consisting of contractible, aligned skeletal myocytes driven by microslotted films. The C2C12 skeletal myocytes were better aligned and differentiated when cultured on styrene-block-butadiene-block-styrene (SBS) microgroove membranes. Electrical stimulation was applied to the self-standing biohybrid film to trigger its contraction and thus its displacement. The maximum displacement of the film (10 μm × 3 μm × 2.5 μm) was 276 ± 55 μm under electrical stimulation at 1 Hz (40 V, 20 ms pulse width). The obtained results were asymptotically compatible with the results of finite element simulation. It is worth mentioning that the study in question laid the foundation for predicting the shrinkage properties of elastic films, highlighting the potential of micro grooved SBS films as an ultraflexible platform for biohybrid machines (Figure 6B) [137]. The advantages of bioinert polymers make them very suitable as extracellular materials in myocyte-driven robots, which can be safely and effectively integrated with biological components.

### 3.2. Hydrogel

The main difference between bioactive hydrogels and artificial hydrogels lies in their origin, composition, and properties. Bioactive hydrogels are composed of biological macromolecules, which have better biocompatibility and bioactivity and can easily be cultured in cells; artificial hydrogels are composed of synthetic polymers, which have better mechanical properties, chemical stability, and controllability. However, their biocompatibility, biodegradability, and bioactivity may be inferior to that of bioactive hydrogels.

#### 3.2.1. Bioactive Hydrogels

Bioactive hydrogels, such as fibrinogen and gelatin derivatives, are usually derived from biological organisms (animals or plants). They therefore have similarities to an extracellular matrix (ECM), i.e., they have similar water content and exhibit mechanical properties similar to those of natural tissues [138]. Their availability is significantly better; thus, they have been widely used for the fabrication of myocyte-driven robots. Xu et al. used FN micrographs as adhesion guides to control the growth of cardiomyocytes, causing them to differentiate and form muscle tissue, which was then combined with a flexible scaffold and conversion system to make a swimming robot. The robotic system purposefully alleviated defects such as electrolysis, allowing it to survive for up to 3 weeks [61]. Gelatin methacrylate (GelMA) hydrogels are a common bioactive hydrogel. Shang et al. cultured cardiomyocytes on GelMA hydrogels and ovoid macroporous anisotropic inverse opal substrates to fabricate a composite biohybrid actuator with self-driving ability and self-reported feedback. Cardiomyocytes were induced by the anisotropic surface morphology of the hydrogel with a highly oriented and ordered layout. The driver could produce color changes through periodic contractions and enabled visual feedback functions that allowed real-time observations of the driver’s operating state (Figure 6C) [139]. Shin et al. developed a bio-inspired soft robotic system with a model consisting of two different micropatterns of PEG and CNT-GelMA hydrogel layers. Cardiomyocytes were seeded on CNT-GelMA hydrogel patterns covered with PEG-patterned hydrogels for mechanical stabilization. Gold microelectrodes were added between the two materials to control the beating behavior of the bionic soft robot using a flexible gold microelectrode array for local stimulation (Figure 6D) [73].

#### 3.2.2. Artificial Hydrogels

Although artificial hydrogels offer superior batch performance and consistency and lower costs compared to naturally derived bioactive hydrogels, they lack cell adhesion. To address this issue, they are either modified with bioactive patterns or used as biologically inert carriers. Common artificial hydrogels include polyvinylidene fluoride (PVDF), poly(3,4-ethylenedioxythiophene) (PEG), poly(3,4-ethylenedioxythiophene) (PEDOT), etc. Yoon et al. designed and constructed compact and miniaturized biohybrid microcolumns in response to the spontaneous contraction of neonatal rat cardiomyocytes. One side of the microcolumn was coated with bioactive FN for the attachment of neonatal rat cardiomyocytes, and the other side was coated with bioinert PEG to avoid cell attachment. As a result, the cardiomyocytes grew unevenly on one side of the cylinder and showed a significant bending effect along the microcolumn. The force generated by the contraction of the cardiomyocytes was 191 μN [140]. Kim et al. cultured C2C12 skeletal myocytes on a multiwalled carbon nanotube (MWCNT)-functionalized PEDOT surface to achieve bionic drive. The worm robot (60 mm long and 10 mm wide) could rhythmically contract and relax through the application of a periodic voltage to the ribbon driver. Furthermore, its maximum contraction length could be adjusted between 0.2 and 0.7 mm by setting the period of applied voltage (Figure 6E) [72]. Liu et al. proposed a piezoelectric nanogenerator. The nanofibers were driven by cardiomyocytes embedded in a fibrinogen-modified polyvinylidene fluoride (PVDF-FN) hydrogel. A current output of 45 nA and a voltage output of 200 mV were generated under spontaneous beating at 1.1 Hz [141].

### 3.3. Tissue-Harvested Biomaterials

The biomaterials most relevant to myocyte-driven robots are tissue-harvested biomaterials (e.g., decellularized extracellular matrix, collagen, etc.). Despite their high cost and low availability, they are still widely used in the fabrication of myocyte-driven robots. Webster et al. used primary cardiomyocytes or primary skeletal myocytes isolated from chicken embryos as drivers and then inoculated the cells into a scaffold composed of electrochemically aligned collagen (ELAC) to make two myocyte drivers. The micropatterned matrix promoted cell adhesion and induced cell alignment. Under the same test conditions, the average velocity of the skeletal myocyte driver was approximately 77.6 ± 86.4 μm/min, while the average velocity of the cardiomyocyte driver was 9.34 ± 6.69 μm/min. The performance of the ELAC scaffold could be altered by changing the compaction time and the cross-linking rate, which, in turn, allowed the optimization of the scaffold’s geometry to improve the device’s performance [142]. Pagan-Diaz et al. mixed skeletal myocytes with the extracellular matrix (ECM) proteins fibrinogen and thrombin, and the mixture was injected into PEGDA molds and cultured to form muscle tissue. The muscle tissue was then combined with the PEGDA skeleton to make a myocyte-driven walking robot. A platform was designed by linking the computational modeling of the developed biorobot with empirical validation. The platform not only addressed the burden of thickness due to limited nutrient diffusion (resulting in an extension of the thickness of 1.3 cm), but also guided the optimization of the force output of the autonomous hybrid skeletal muscle robot from 200 μN to 1.2 mN (Figure 6F) [65].

In summary, the choice or combination of substrates for myocyte-driven robots usually depends on the end goal. The substrate materials and their macro-and microgeometric designs can jointly influence the driving force and power density of biorobots. The combination of multiple materials to synthesize composite multifunctional materials may be a new trend for future myocyte-driven robots.

**Figure 6 micromachines-14-01643-f006:**
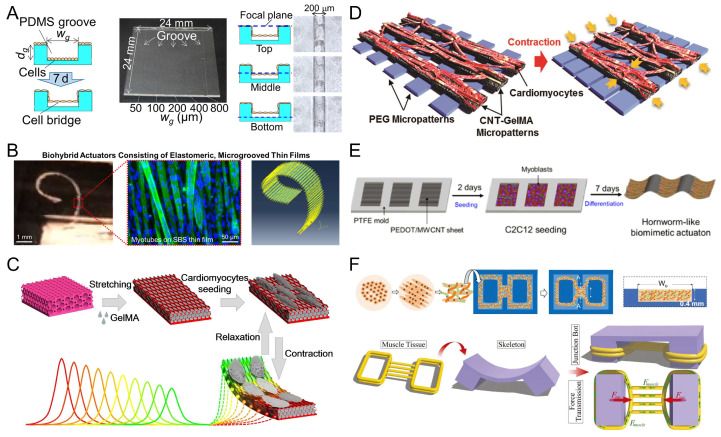
Examples of myocyte-driven robotic extracellular materials section. (**A**) Formation of PDMS microgroove structures for cardiomyocyte microtissue bridging walls. Reproduced from Reference [86] with permission from Sensors and Actuators B: Chemical. (**B**) A biohybrid device formed by culturing C2C12 skeletal myocytes on SBS microgroove membranes. Reproduced from Reference [137] with permission from ACS Biomater Sci Eng. (**C**) A bio-drive consisting of cardiomyocytes cultured in GelMA hydrogels. Reproduced from Reference [139] with permission from ACS Nano. (**D**) Formation of a bio-drive on CNT-GelMA and PEG hydrogels seeded with neonatal rat cardiomyocytes. Reproduced from Reference [73] with permission from Adv Mater. (**E**) PEDOT/MWCNT sheets with skeletal myocytes cultured to form a bio-drive. Reproduced from Reference [72] with permission from Scientific Reports. (**F**) Skeletal myocytes were implanted into a model consisting of Matrigel, thrombin, and fibrinogen to form muscle tissue. Reproduced from Reference [65] with permission from Advanced Functional Materials.

## 4. Contraction of Muscle Tissue and Control Methods

The main driving mechanism of myocyte-driven robots is the deformation of the flexible substrate by muscle contraction. The contractility of the muscle tissue is critical to the design and preparation of robots [124,143]. The motion of myocyte-driven robots can be driven by the spontaneous contraction of muscle tissue. The robot can also be driven by external stimuli to achieve different forms of motion, allowing controllable motion of the robot.

### 4.1. Spontaneous Contraction

Since cardiomyocytes can spontaneously contract, most cardiomyocyte-driven robots utilize the spontaneous contraction of cardiomyocytes to drive the robot’s operation. Xi et al. first grew and self-assembled individual cardiomyocytes into muscle bundles, and then integrated the muscle bundles with micromechanical structures to form a self-assembled microdevice driven by cardiac muscle [40]. The spontaneous contraction of the cardiomyocytes allowed them to act as the actuators of the pump’s body, using the endogenous ability of the cells to convert chemical energy into mechanical energy. Tanaka et al. fabricated an on-chip actuated pump driven by cardiomyocytes. The cultured cardiomyocyte slice was coupled to a PDMS microchip with a microchannel. A pusher mechanism was installed between the cardiomyocyte slice and the septum in the microchannel, and the pusher mechanism transferred the contraction force of the cell slice into the fluid (Figure 7A) [83]. Michas et al. combined cultured cardiac muscle tissue with a small metamaterial scaffold fabricated using two-photon direct laser writing to fabricate a microfluidic ventricle driven by cardiomyocytes. Compression of the helical scaffold by contraction of the heart tissue produced fluid flow in the channels of the device, recreating the flow control function of the human ventricle [16]. Similarly, the spontaneous contraction of cardiomyocytes can be used to create locomotor robots with motor functions such as walking, swimming, etc. Kim et al. fabricated a crab-like microrobot, which had an asymmetric structure with three front and hind legs of different lengths. When the cardiomyocytes on the surface of the legs begin to beat synchronously, the three legs of the robot showed vertical displacement due to the contractile force of the cardiomyocytes, which drove the miniature bionic robot to move [78]. Chan et al. fabricated a cardiomyocyte-driven walking robot in which cardiomyocytes from a neonatal rat heart were extracted and implanted into a cantilever beam. The cardiomyocytes attached to the cantilever increased in size and exhibited spontaneous contractile activity. Retraction of the cantilever’s structure by the contractile force of the cardiomyocyte sheets drove the robot’s motion (Figure 7C) [79].

Under certain conditions, skeletal muscles can likewise produce self-stimulation, causing the muscle to contract. Cells can sense and respond to mechanical stimuli and translate them into intracellular biochemical responses, an ability known as mechanotransduction [144]. Guix et al. developed a skeletal muscle-based swimming robot that transferred a scaffold loaded with skeletal myocytes into a serpentine spring skeleton, which produced cell maturation while allowing mechanical integrity and self-stimulation. The inherent resilience of the spring system allowed the skeletal muscles to dynamically comply during spontaneous contractions, constituting a cyclic mechanical stimulus to the extent that the spontaneous contractions of the skeletal muscles were used to drive the robot’s motion in the absence of external stimuli [60]. However, such robots rely solely on spontaneous movement of the muscle tissue, and they lack control and versatility.

### 4.2. Electrical Stimulation

Electrical stimulation (ES) is one of the main methods of controlling myocyte-driven robots. ES has significant effects on the cells’ alignment and synchronized beating and can thus control the frequency of contraction and the force of the myocytes or tissues [122,145,146]. The effects of ES on myocytes’ alignment and differentiation are critical because they are important factors in the manufacture of functional muscle myofibers. ES can propagate through thick myocyte structures or muscle myofibers and it has a significant effect on the alignment of C2C12 myotubes [113]. In living organisms, muscle contraction and relaxation depend on bioelectrical transmission. In particular, skeletal muscles are stimulated in vivo by the nerves to produce contractions, and ES can mimic the activity of motor neurons on skeletal muscle tissue. ES causes the depolarization of the myocytes’ membrane, which triggers excitation–contraction coupling [147]. Morimoto et al. proposed a biohybrid robot driven by a pair of antagonistic skeletal muscle tissues. The contraction of antagonistic skeletal muscle tissues was controlled by ES to achieve the function of manipulating objects. Moreover, the range of the rotation angles of the joints and the strain of the skeletal muscle tissue could be changed by controlling the frequency and magnitude of the electric field [43]. Pagan-Diaz et al. anchored skeletal muscle tissue between the two legs of a robot and electrically stimulated the muscles to contract periodically, thereby inducing motion through a double-anchored gait. The walking speed of the robot was proportional to the stimulation frequency [65]. In addition, electrical stimulation of the contraction of skeletal muscle has been applied in pump robots. Li et al. designed and developed a biohybrid impedance pump in which a hydrogel hose was connected to a rigid tube at both ends to form a closed loop, and a skeletal muscle ring was used to wrap the hydrogel hose. Multiple myotubes of the skeletal muscle ring were induced by ES to interact in a coordinated manner, thereby enhancing the contraction force and squeezing the hydrogel tube to drive the flow of fluid (Figure 8A) [59].

Cardiomyocytes respond with better contractile properties when subjected to ES, which is important for the fabrication of cardiomyocyte-driven robots that require control. Shin et al. proposed an electrically driven microengineered bionic soft-body robot in which cardiomyocytes were seeded in a bionic batfish structure embedded with microelectrodes, which caused nonspontaneous excitation of the cardiomyocytes under stimulation by microelectrodes. The electrical conduction through the cardiomyocytes caused a wavelike displacement of the entire structure, whose motion originated in the center of the body and propagated to the outer fin region, mimicking the physiological drive of the muscles. It is worth mentioning that when the robot was stimulated at frequencies higher than 0.5 Hz, the robot did not have enough time to complete a complete contraction cycle, reducing the drive amplitude of the robot. Thus, the robot’s drive did not precisely follow the applied frequency; however, it was still controllable up to 2.0 Hz (Figure 8B) [73].

Although ES is suitable for the control of myocyte-driven robots, it also has certain disadvantages. For example, ES causes the electrolysis of the culture medium and produces substances that are harmful to the cells. To improve the differentiation of adult myocytes, Liu et al. proposed a ring-distributed multielectrode (CEs) method containing 12 electrodes, which was compared with conventional parallel electrodes (PEs) through a simulation. The results showed that PEs produced a much larger area of localized electric fields with high intensity than CEs, which may damage cells. Moreover, CEs could induce the differentiation and secondary formation of C2C12 myotubes more effectively than PEs, and the CEs stimulated the cells with better differentiation, myotube length, myotube width, and myotube alignment. The proposed CEs further optimized the method of electrical stimulating myocytes (Figure 8C) [148]. Zhang et al. proposed a manta ray biomimetic fusion robot controlled dynamically by circularly distributed multielectrodes (CDME), which controlled the direction of the electric field generated by the CDME in real time, parallel to the muscle-driven tissue of the dynamic swimmer, and applied dynamic electrical stimulation to the muscle tissue, ensuring the stable controllability of the muscle-driven robotic swimming. The rotational electrical stimulation of CDME was used to cultivate skeletal muscle rings, which facilitated cell differentiation and enhanced the driving force of the robot’s muscle tissue (Figure 8D) [15].

### 4.3. Optical Stimulation

Optical stimulation control of myocyte-driven robots is usually guided by light to achieve complex movements through optogenetic cell-driven robots. Optogenetics combines light and genetics to precisely control living cells with tailored functions [149]. Optogenetics transfers genetic information from photosensitive proteins to target cells by viral transfection, transgenic animals, etc., and transfers photosensitive genes (e.g., ChR2, NpHR, Arch, OptoXR, etc.) into specific types of cells in the nervous system for the expression of specific ion channels or GPCRs. Photoreceptor ion channels are selective for the passage of cations or anions, such as Ca^2+^, K^+^, etc., under different wavelengths of optical stimulation, thus causing changes in the membrane potential on both sides of the cell membrane to achieve selective excitation or inhibition of the cell [150,151,152]. Optogenetics has the advantages of noninvasiveness, rapid response, adjustable reversibility, and high spatial resolution.

Optogenetics allows for the user-defined spatiotemporal activation of muscle actuators by genetically programming the cells to express photosensitive proteins [116,153]. Bruegmann et al. sustained muscle tissue contraction by linking the repetitive frequency of optical stimulation to effective depolarization and repolarization of the membrane potential [154]. Thus, through the use of optogenetics and optical stimulation, muscle tissue can be precisely controlled, which, in turn, can enable the control of myocyte-driven robots. Raman et al. developed myocyte-driven robots using an optogenetic skeletal muscle tissue drive system by using an existing lentiviral transduction scheme to modify C2C12 myogenic cells from mice with a mutant variant of the blue light-sensitive ion channel, channel retinoid-2 (ChR2). The cell solution was injected into skeletal muscle rings in molds and then the muscle rings were transferred to 3D-printed hydrogel robotic skeletons. The muscle rings were contracted by optical stimulation to drive the robot and achieve directional motion with an average velocity of 310 μm/s and two-dimensional rotational steering with an average rotational velocity of 2°/s (Figure 9A) [81]. Similarly, cardiomyocytes can be reprogrammed to have light-responsive characteristics to achieve coordinated action and complex functional behavior. Park et al. combined optogenetic cardiomyocyte tissue with soft materials to create an artificial skate that contained a microfabricated gold skeleton, in which the movement was controlled by optical stimulation. The cardiomyocytes were given the ability to respond to optical stimulation by modifying them to express the channel retinoid-2 (ChR2). Under periodic optical stimulation, a forward thrust was generated by the fluctuating motion of the fins to make the artificial skate move forward rhythmically and continuously. When the gait and turn of the ray were controlled, it could swim over obstacles at a faster speed (Figure 9B) [42]. Lee et al. constructed a biohybrid fish equipped with an antagonistic muscle bilayer and a geometrically insulated heart tissue node (G-node). The G-node acted as a pacemaker-like autonomic pacing node that regulated the frequency and rhythm of spontaneous cardiomyocyte contractions. The muscle bilayer and the autonomic pacing node generated continuous, spontaneous, and coordinated back-and-forth fin movements that drove the fish forward. Blue light-sensitive (ChR2) and red light-sensitive (ChrimsonR) ion channels were expressed in both muscle layers using lentiviral transduction. Precise control of the robot fish’s swimming ability was achieved by alternately stimulating the muscle bilayer with blue and red light pulses, causing the muscle layers on both sides to contract alternately (Figure 9C) [45]. Aydin et al. proposed a neuromuscular unit-driven swimming robot. The body of the robot consisted of a separate soft scaffold, skeletal muscle tissue, and a cluster of optogenetic stem-cell-derived nerves containing motor neurons. Clusters of neural cells with photosensitive ChR2 ion channels and motor neurons expressing Hb9-GFP were obtained by the directed differentiation of optogenetic mouse embryonic stem cells. Optical stimulation of the motor neurons caused periodic contractions of muscle tissue that propelled the flagellum in a temporally irreversible manner, thereby propelling the swimmer forward. The ability to control muscle activity through motor neurons paves the way for the further integration of neural units into biohybrid systems (Figure 9D) [44].

Although optical stimulation has the advantages of fast response and high spatial and temporal resolution and is widely used in the development and study of myocyte-driven robots, it also has some drawbacks that limit the application of optical control. For example, target cells or tissue regions subjected to prolonged light exposure may lead to heating effects, tissue damage, and off-target cellular activity, and continuous light exposure can affect cell viability [155,156]. Light sources such as UV light may damage DNA and proteins in cells or microorganisms. Therefore, the optical stimulation of certain specific light sources should be limited to short exposure times when optical stimulation is used to control robots [157]. Moreover, some ambient light cannot penetrate, largely limiting the range of applications for light-controlled myocyte-driven robots.

### 4.4. Chemical Stimulation

Many compounds have been found to have a significant effect on the contraction frequency and contractile force of myocytes. The beat frequency of myocytes can be increased or inhibited by pharmacological stimulation. Epinephrine, for example, can cause a significant increase in the amplitude and frequency of cardiomyocyte contraction, whereas nifedipine can suspend their beats [158,159,160]. Similarly, some compounds such as potassium chloride, caffeine, glutamate, and acetylcholine can induce contraction in skeletal myocytes [161,162,163].

Takemura et al. designed a jellyfish robot composed of cardiomyocyte gel. The cardiomyocyte gel is a tissue-engineered muscle made of rat cardiomyocytes mixed with collagen gel. After chemical stimulation of the jellyfish, it was found that after epinephrine stimulation, the robot’s pulsation increased 2.4-fold compared to pre-stimulation. After the nifedipine application, the pulsation frequency of the robot decreased, and when nifedipine was applied for 4 min, the pulsation of the robot stopped completely. Thus, the robot’s movement speed and start-stop can be controlled by chemical stimulation [76]. A heart chip based on the scaled wings of the great blue flicker butterfly was designed by Chen et al. Engineered cardiomyocytes were cultured on modified natural great blue flicker butterfly wings to construct a biosensor system. The cardiomyocytes would orient themselves along the parallel nanostructures of the butterfly wings and resume beating autonomously. The contraction and diastole of the cardiomyocytes during the periodic beating drove the butterfly wings to undergo synchronous bending deformation, resulting in corresponding changes in their structural color and photonic band gap, leading to the self-reporting of myocardial mechanical properties. With the addition of isoprenaline stimulation, the butterfly wings flapped faster and deformed to a greater extent. This biohybrid system has great potential for biological research and drug development [164].

## 5. Various Applications of Myocyte-Driven Robots

Scientists have been drawing inspiration from nature to develop functional myocyte-driven robots through the application of bionics. Bionics, a significant research approach in robotics, enhances kinematic performance and functionality by emulating the structures and behaviors of natural organisms. Concurrently, myocyte-driven robots, which incorporate natural muscle materials as core components, hold great promise in propelling the field of robotics to new heights [165,166]. Major applications of muscle cell-driven robots include muscle cell-driven swimmers, walkers, grippers, and pump robots. At the same time, it can also be used as a carrier or as tweezers or micromanipulators, biosensors, drug delivery systems, etc.

### 5.1. Swimmers

By mimicking natural organisms (e.g., fish, jellyfish, etc.), myocyte-driven robots have the morphology and locomotor posture of real aquatic organisms. Feinberg et al. fabricated a simple swimmer by culturing neonatal rat ventricular cardiomyocytes on elastomeric membranes (MTFs). The synchronized contraction of the cardiomyocytes bent the film and returned to its original shape during diastole, thus propelling the robot to swim forward. The robot obtained a maximum speed of 24 mm/min in the anisotropic mode of the MTFs [122]. Xu et al. fabricated a soft-bodied robot driven by the interaction between a flexible membrane in the tail and a liquid solution. The membrane effectively acted as a muscular “tail fin” through the micropatterning of cardiomyocytes on the flexible membrane. The synchronized contraction and relaxation of the cardiomyocytes allowed the membrane to move up and down in the fluid, propelling the robot forward. The robot could be remotely controlled by NIR stimulation based on its deformable mechanical structure. Under NIR irradiation, the support plate absorbed the NIR energy and curled, which, in turn, caused the wing to contract, thus inhibiting its motion and stopping the robot completely (Figure 10A) [61]. Zhang et al. developed a swimming robot driven by skeletal muscle tissue cultured in vitro and controlled by CDME, using manta rays as the inspiration for its design. Driven by skeletal muscle tissue, effective propulsion of the robot was achieved. Dynamic control was achieved by controlling the directionally controllable stimulation electric field of the CDME. Its swimming speed reached 70 μm/s [15]. However, a limited lifespan and power are problems and challenges still facing the development of myocyte-driven swimmers, which may be solved through the advancement of technology.

### 5.2. Walkers

Walking or crawling is another common type of movement in myocyte-driven robots. The spontaneous contraction or stimulus-induced contraction of myocytes is used to drive the robot forward by causing the flexible substrate to undergo anisotropic deformation. Kim et al. developed a crab-like microrobot including cardiomyocytes and elastic material that showed prolonged actuation under physiological conditions. The microrobot was able to walk continuously for more than 10 days and travel an estimated total distance of 50 m in a week at an average speed of 100 μm/s [78]. Inspired by the crawling mechanism of snakes and caterpillars in nature, Sun et al. proposed a myocyte-driven robot consisting of asymmetric claws and carbon nanotube (CNT)-induced cardiac tissue layers. The asymmetric claws served as a support point to provide frictional force, which enabled the whole soft-body robot to accomplish directional motion during the contraction of the myocytes. These three functional layers allowed the robot to closely mimic the crawling behavior of caterpillars. The robot was stimulated by different concentrations of drugs and exhibited different motility velocities, with a maximum speed of 20 μm/s (Figure 10B) [66]. Skeletal myocytes are an alternative to myocyte-driven power sources for walkers. Skeletal muscle tissue was wrapped around a hydrogel column to mimic the muscle–tendon–skeleton mechanism in animals. Cvetkovic et al. proposed a 3D biorobot with symmetrical and asymmetrical pillar structures driven by skeletal muscles. The contraction of the skeletal muscle tissue was triggered by ES to drive the robot and control its direction of motion. The maximum velocity of the robot with an asymmetric strut design when crawling was 156 μm/s [80]. Wang et al. proposed a dual-ring biorobot consisting of two independent skeletal muscle ring drivers and a tetrapod hydrogel skeleton with anterior–posterior asymmetry. Skeletal muscle tissue was combined with differential electrical stimulation to drive the robot’s motion. After the study, it was found that the biorobot with high passive force walked at an average speed of 2.5 mm/min; in contrast, the biorobot with low passive power had more than twice the speed of the former, with an average speed of 5.9 mm/min. The robot could be rotated by differential stimulation of the muscle tissue on both sides to break the left–right driving symmetry and generate net lateral friction [82]. However, the speed of walking or crawling robots is limited compared with myocyte-driven swimming robots, and their performance needs to be further improved.

### 5.3. Grippers

Soft gripper robots can manipulate objects smoothly and directly without deformation. In recent years, myocyte-driven grippers have been gaining ground due to their natural biocompatibility, high energy conversion efficiency, and soft properties. Hoshino and Morishima combined body hair from dogs with primary skeletal muscles from rats to fabricate a muscle-powered microtweezer cantilever. The cantilever used a hair as a skeleton, and ES caused the myotubes to contract, driving the cantilever to produce displacement, simulating a grasping function. However, its displacement was not sufficient for effective grasping [167]. Kabumoto et al. constructed a micromanipulator by mixing skeletal cells from rats with collagen gel to form a skeletal muscle gel. The skeletal muscle gel contracted under ES, which, in turn, controlled the displacement of the micromanipulator. The microhand’s displacement was about 8 µm, which could be used to manipulate objects with a size of about 200 µm [67]. Morimoto et al. proposed a manipulator consisting of a rigid skeleton and a pair of antagonistic skeletal muscle tissues. The skeleton had electrodes capable of providing stimulation to induce the contraction of the muscles, a rotatable joint, and a flexible band that could drive the joint to rotate. The cultivated skeletal muscle tissue was fixed on the robot’s skeleton to form a manipulator with antagonistic skeletal muscle tissue. The joints could be rotated up to 90° by selectively contracting the skeletal muscle tissue. Experiments have shown that a single manipulator could complete the operation of picking up and placing a circle, and the coordinated operation of two manipulators could complete the action of picking up a box. Notably, the manipulator was long-lived, retaining its initial contractility after one week (Figure 10C) [43]. In 2020, the group proposed a similar skeletal muscle-driven manipulator. Skeletal muscle tissue was encapsulated in collagen to obtain the necessary humidity to allow the manipulator to move through the air through contraction of the skeletal muscle tissue. The applicability of the manipulator has been further improved [52]. However, gripper technology has been less studied in myocyte-driven robotic devices compared with other types of robots. Although great progress has been made in the control methods and the corresponding movement patterns, continued development is needed.

### 5.4. Pump-Bots

Due to their ability to contract spontaneously, cardiomyocytes can be used as actuators to create an automatic biological pump. Cardiomyocytes can be driven using the cell’s endogenous ability to convert chemical energy in the culture environment into mechanical energy [83]. Tanaka et al. created a microspherical heart pump driven by a cardiomyocyte slice. Cardiomyocyte sheets were obtained by culturing them on a temperature-responsive PIPAA layer and then wrapped around a flexible hollow sphere with inlet and outlet ports fabricated from PDMS. Synchronous pulsation of the cardiomyocytes induced oscillation of the fluid within the capillaries connecting the hollow spheres at a flow rate of 0.047 mL/min, allowing continuous pumping for 5 days [85]. However, the process of separating the cardiomyocyte sheets from the culture matrix and transferring them to soft materials has certain difficulties. Park et al. overcame these limitations by fabricating a mock biomixing pump with a dome structure. A noninvasive method was used to implant cardiomyocytes on dome-shaped PDMS films with a Cr/Au layer. The contractile force generated by the cardiomyocytes cultured on the membrane moved the membrane up and down, causing the contraction and relaxation of the microchambers, which, in turn, led to the flow of fluid in the microchannel [84]. Tanaka et al. developed an ultrasmall cardiomyocyte-driven autonomous hybrid pump with dimensions of only 200 μm × 200 μm × 150 μm. The cardiomyocytes self-organized into microtissues connecting the walls of the PDMS microchannel structure, forming several contractile bridges. The micropump displayed spontaneous and periodic oscillations with a theoretical flow rate of 1.0 nL/min [86]. Skeletal myocytes can also act as drivers. Unlike the spontaneous contraction of cardiomyocytes, skeletal myocytes have the advantage of being able to be precisely controlled. Li et al. developed a valveless pump robot powered by engineered skeletal muscles. The pump robot consisted of a soft hydrogel tube connected at both ends to a stiffer PDMS holder, thereby creating an impedance mismatch. A muscle ring was wrapped around the hydrogel tube at an off-center location. The asymmetric placement of the muscle ring resulted in wave generation at a temporal interval, which resulted in unidirectional fluid flow. The pump robot achieved flow rates of up to 22.5 μL/min (Figure 10D) [59]. In 2022, the group reported a feedback loop pumping system driven by engineered skeletal muscles. A muscle ring was made from skeletal myocytes and a Type I collagen/Matrigel matrix, which was then combined with a soft hydrogel tube attached to a rigid fluid platform at both ends. The muscle ring contracted in a repetitive manner, autonomously squeezing the tube, thereby creating an impedance pump. The resulting flow circulated back to the muscle ring, forming a feedback loop that allowed the pump to respond to the flow it generated and adaptively manage its pumping performance based on the feedback. Its static flow rate was 13.62 μL/min [69]. However, the muscle-driven pump robot had some problems during practical application, such as increasing the flow rate, stability, and long-term use, which need further improvement.

## 6. Summary and Perspectives

The future perspectives of myocyte-driven robots are promising as they have the potential to overcome the limitations of conventional artificial drive technology and replace conventional drives. With the development of cell engineering, micro- and nanotechnology, and 3D bioprinting, myocyte-driven robots can offer self-assembly, self-healing, multiple degrees of freedom, and intelligent sensing while providing superior controllability and output force compared with conventional rigid robots.

The integration of myocytes with biocompatible materials in microfluidic devices to form organ-on-a-chip systems represents a promising avenue for drug testing and disease monitoring [168]. The beat frequency of cardiomyocytes changes in response to different drug concentrations and structural color indicators can dynamically reflect the state of movement of a bionic robot [164]. To accommodate actual applications in clinical practice, future microfluidic platforms should have more robust and complex capabilities. When a robot needs to enter the human body for surgery (especially in complex and small spaces such as blood vessels), problems in terms of controllability, physical compatibility, and degradability of the microrobot arise. Myocyte-driven robots are miniaturized, highly biocompatible, and biodegradable. They can be used to access the circulatory system for monitoring disease or to perform minimally invasive surgery. After completing its task, the robot can safely self-degrade in the body. With continuous improvement in the methods of controlling myocyte-driven robots and their functions, targeted surgical treatment of different organs could be performed in the future. The self-assembly and self-healing properties of myocytes could also enable the inspection of environmental corrosiveness with myocyte-driven robots, providing a valuable tool for environmental monitoring [169]. The small size and flexibility of myocyte-driven robots make them suitable for performing tasks in challenging environments, for example, battlefield reconnaissance, disaster rescue, and deep-sea exploration. However, there are still many challenges for myocyte-driven robots, as follows:

1. The ethical issues of primary myocytes. Primary myocytes need to be obtained from living organisms and therefore present an ethical problem. To address this issue, different methods of differentiation and induction into myocytes are available [170,171]. However, this approach has limited opposition relative to the use of natural sources, and the cost of obtaining such myocytes in large numbers is more expensive.

2. The nutrient delivery and survival environment of myocytes is a practical difficulty. Specific culture conditions (e.g., medium composition, temperature, pH) are required for survival. Nutrients and O2 must be regularly delivered. Currently, myocyte-driven robotic motion compatible with ambient air has been achieved by methods such as wrapping collagen around the muscle tissue. Moreover, the external stimuli applied to muscle tissue in order to control the robot can damage the myocytes and lead to a decrease in the robot’s performance. Its lifespan and structure still need further optimization.

3. Myocyte-driven robots lack intelligent perception of the external environment. Their power source comes from the contraction of muscle tissue; thus, they can be made to move using the spontaneous contraction of muscles or through external stimulation. Currently, myocyte-driven robots are mostly controlled by physical or chemical stimulation, while intelligent perception and control functions have been less frequently studied. In the future, myocyte-driven robots may respond to external signals through an intelligent sensing system, thus realizing a closed-loop feedback control system.

4. The structures of myocyte-driven robots are generally simple, have limited degrees of freedom, and are relatively homogeneous in function. Most robots can only be controlled as a whole. It may be possible to adopt the co-culture of different cells in combination with composite multifunctional materials or achieve the independent control of specific parts of the muscle tissue of myocyte-driven robots.

5. There is complexity in the control of robots. Robots that utilize myocyte drive require precise control of muscle cell contraction and diastole. This may involve complex biological and electrophysiological mechanisms, requiring the use of high-end techniques and equipment that increase the difficulty of control.

6. Myocyte-driven robots have limited power. Compared to conventional robots, myocyte-driven robots have a more natural and fluid power source. The ability of muscle tissue to contract can mimic the way human muscles move, resulting in more natural robot movements. However, myocyte drive has limitations in terms of power and speed.

Myocyte-driven robotics is a rapidly developing multidisciplinary field, which has been reviewed in this paper in four main areas: myoblast types, extracellular materials, control methods, and types of motion. The development and application of myocyte-driven robotics require the cooperation of different fields, and it is a perfect example of cross-fertilization between engineering and life sciences. With cell engineering, micro and nanotechnology, and 3D bioprinting continuing to evolve, myoblast-driven robotics is poised for even greater and more exciting accomplishments. The potential application areas for myocyte-driven robotics are vast, including in vitro drug testing, disease monitoring, minimally invasive surgery, environmental monitoring, and exploration in challenging environments. While there are still many challenges to be solved, the future of myocyte-driven robotics is bright, and continued research and development will lead to many more breakthrough innovations.

## Figures and Tables

**Figure 2 micromachines-14-01643-f002:**
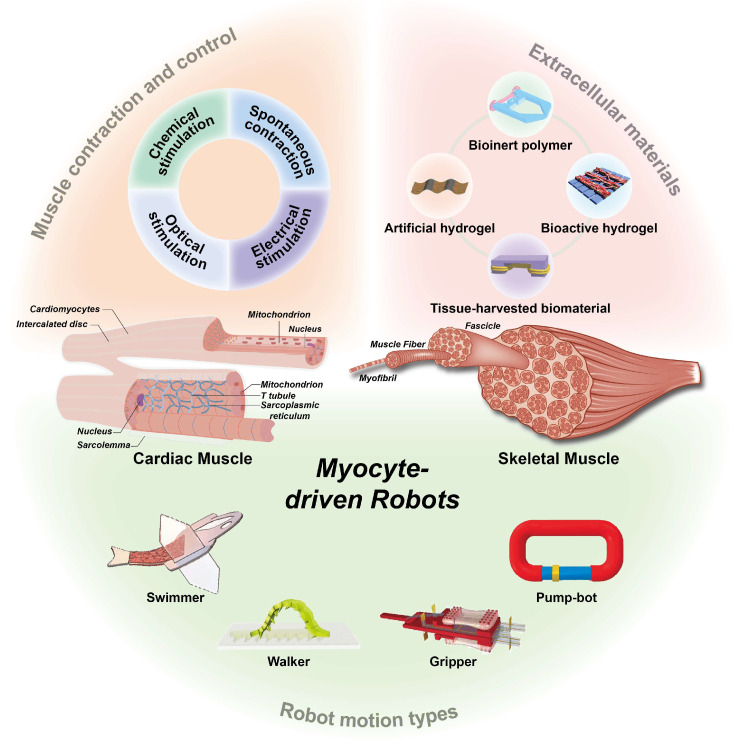
Overview of myocyte-driven robots, including cardiac muscle and skeletal muscle structures, extracellular materials, robot control methods, and types of robot motion. The micro hand was fabricated by PDMS using soft lithography. Reproduced from Reference [67] with permission from 2010 3rd IEEE RAS & EMBS International Conference on Biomedical Robotics and Biomechatronics. PEDOT/MWCNT sheets reproduced from Reference [72] with permission from Scientific Reports. A model consisting of two different micropatterns of PEG and CNT-GelMA hydrogel layers reproduced from Reference [73] with permission from Adv Mater. PEGDA skeleton reproduced from Reference [65] with permission from Advanced Functional Materials. An autonomously swimming biohybrid fish designed with human cardiac biophysics. Reproduced from Reference [45] with permission from Science. Bioinspired soft robotic caterpillar with cardiomyocyte drivers reproduced from Reference [66] with permission from Advanced Functional Materials. Biohybrid robot powered by an antagonistic pair of skeletal muscle tissues. Reproduced from Reference [43] with permission from Science Robotics. A valveless pump robot powered by engineered skeletal muscles. Reproduced from Reference [59] with permission from Proceedings of the National Academy of Sciences.

**Figure 3 micromachines-14-01643-f003:**
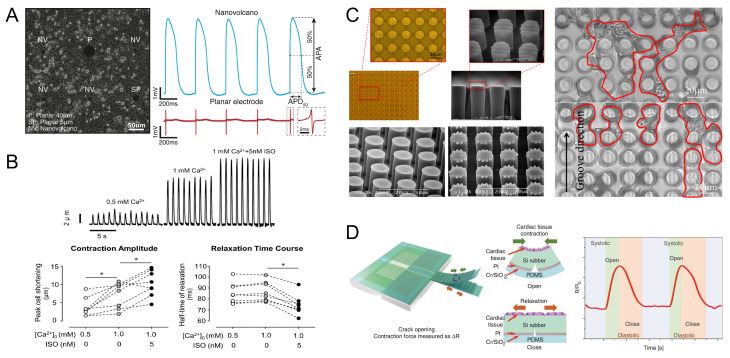
Properties of cardiomyocytes. (**A**) Intracellular recordings of action potentials of cardiomyocytes monolayers. Reproduced from Reference [89] with permission from Nano Letters. (**B**) Effect of Ca^2+^ concentration on cardiomyocytes contraction amplitude. * *p* < 0.05 vs. extracellular 1 mM Ca^2+^ in the absence of ISO (Bonferroni *t*-test).Reproduced from Reference [94] with permission from Comput Biol Med. (**C**) Optical images of cardiomyocytes seeded on a microarray column. Reproduced from Reference [96] with permission from Sensors (Basel). (**D**) Schematic diagram of a silicone rubber cantilever beam composed of different layers and the operating principle of a cantilever beam sensor-integrated pdms-encapsulated crack sensor to measure cardiomyocytes contraction in fluid systolic force. Reproduced from Reference [97] with permission from Nature Communications.

**Figure 5 micromachines-14-01643-f005:**
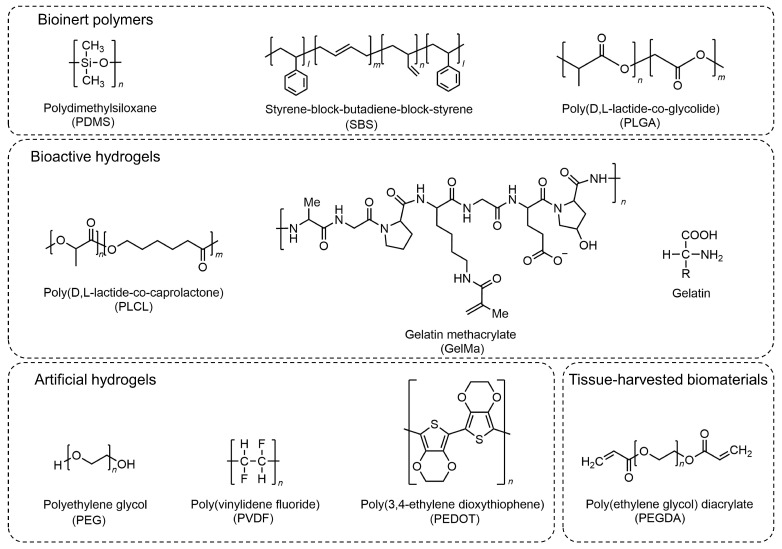
Chemical structures of extracellular materials commonly used in myocyte-driven robots, including their corresponding abbreviations.

**Figure 7 micromachines-14-01643-f007:**
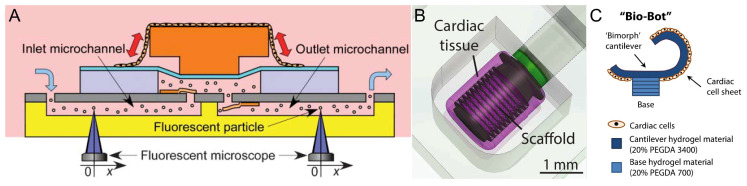
Examples of robots driven by spontaneous contraction of myocytes. (**A**) A cellular micropump chip was constructed using a cardiomyocyte membrane sheet as a prototype. Reproduced from Reference [83] with permission from Lab on a Chip. (**B**) A live heart pump was fabricated on a chip using high-precision fabrication techniques. Reproduced from Reference [16] with permission from Science Advances. (**C**) Cardiomyocyte-driven walking robot. Reproduced from Reference [79] with permission from Sci Rep.

**Figure 8 micromachines-14-01643-f008:**
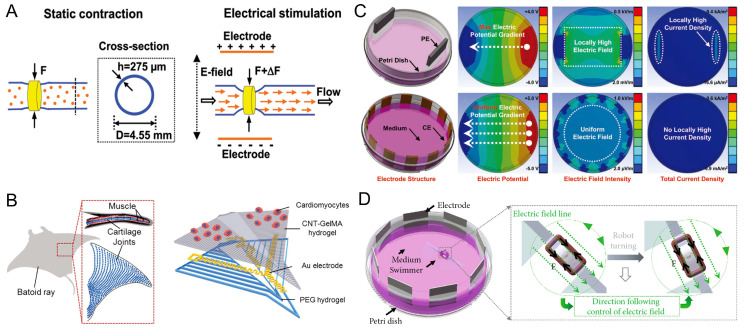
Examples of electrically stimulated, controlled, myocyte-driven robots (**A**) Schematic diagram of a skeletal muscle-driven pump deformed by static muscle contraction and electrical stimulation. Reproduced from Reference [59] with permission from Proceedings of the National Academy of Sciences. (**B**) Electrically driven micro-engineered bionic soft robot. Reproduced from Reference [73] with permission from Adv Mater. (**C**) Simulation results of electrical properties of Pes with CEs. Reproduced from Reference [148] with permission from Soft Robot. (**D**) A manta ray bionic fusion robot dynamically controlled by circularly distributed multi-electrodes (CDME). Reproduced from Reference [15] with permission from Cyborg and Bionic Systems.

**Figure 9 micromachines-14-01643-f009:**
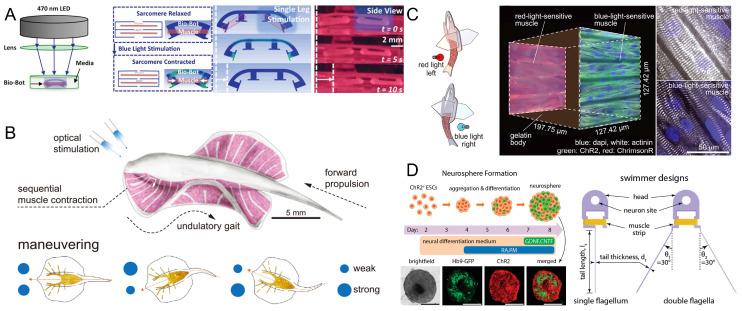
Examples of optical stimulation-controlled myocyte-driven robots. (**A**) An adaptive biological machine driven by optogenetic skeletal muscles. Reproduced from Reference [81] with permission from Proc Natl Acad Sci USA. (**B**) An artificial skate with movement controlled by optical stimulation. Reproduced from Reference [42] with permission from Science. (**C**) A biological hybrid fish with precise control achieved by optical stimulation. Reproduced from Reference [45] with permission from Science. (**D**) A swimming robot is driven by neuromuscular units. Reproduced from Reference [44] with permission from Proc Natl Acad Sci USA.

**Figure 10 micromachines-14-01643-f010:**
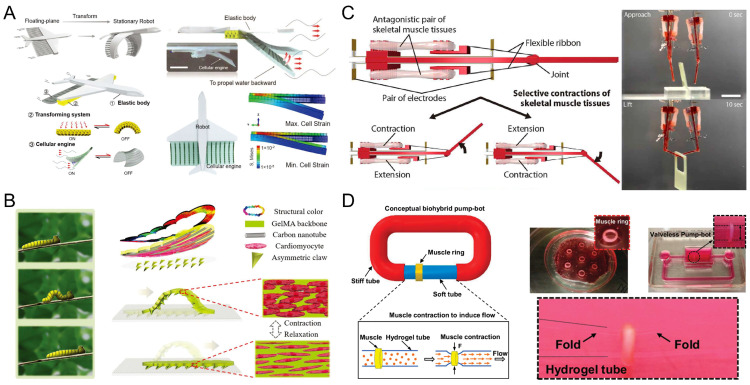
Myocyte-driven robots with different types of motion. (**A**) A remotely controlled deformable swimming robot based on an engineered heart tissue structure. Reproduced from Reference [61] with permission from Small. (**B**) A bionic soft robot caterpillar with a cardiomyocyte driver. Reproduced from Reference [66] with permission from Advanced Functional Materials. (**C**) A manipulator consisting of a rigid skeleton and a pair of antagonistic skeletal muscle tissues. Reproduced from Reference [43] with permission from Science Robotics. (**D**) A valveless pump robot powered by engineered skeletal muscles. Reproduced from Reference [59] with permission from Proceedings of the National Academy of Sciences.

**Table 1 micromachines-14-01643-t001:** Advantages and disadvantages of the three types of robots.

Robot Types	Advantage	Disadvantage	References
Traditional rigid robots	High output power; High speed; High accuracy; Easy manipulation	Complex structure; Less flexible; Poor reliability; Low energy conversion rate	[19,20,21]
Flexible material-driven robots	Light weight; High adaptability to target shapes; High flexibility	Low lifetime; Inefficient movement	[22,23,24,25,26,27]
Biomaterial-driven robots	Excellent biocompatibility; High sensitivity; High stability; High energy conversion rate; Self-assembly and self-healing capability	Low lifetime; Ethical Issues; Cell survival environment issues; Simple function	[36,37,38,39]

**Table 2 micromachines-14-01643-t002:** Design, materials, and performance of myocyte-driven robots with different bionic strategies.

Motion Types	Year	Myocytes	Extracellular Materials	Performance Parameters	Control Methods	References
Swimmers	2011	Cardiomyocytes	Collagen gel	Speed: 6.9 μm/s.	Chemical control	[76]
	2012	Cardiomyocytes	PDMS	Speed: 2.4 mm/s.	Electric control	[41]
	2014	Cardiomyocytes	PDMS	Speed: 81 μm/s.	No control	[63]
	2016	Cardiomyocytes	PDMS; Au	Speed: 1.5 mm/s.	Optical control	[42]
	2016	Cardiomyocytes	PDMS	Speed: 142 μm/s.	No control	[77]
	2018	Cardiomyocytes	PEG;CNT–GelMA Hydrogel; Au	Response time: 0.3 s.	Electric control	[73]
	2019	Cardiomyocytes	FN	Speed: 0.6 ± 0.2 mm/s.	Optical control	[61]
	2019	Skeletal muscles	PDMS	Speed: 0.7 μm/s.	Optical control	[44]
	2021	Skeletal muscles	PDMS	Speed: 800 μm/s.	No control	[60]
	2022	Cardiomyocytes	Gelatin	Speed:15 mm/s.	Optical control	[45]
	2022	Skeletal muscles	PDMS	Speed: 70 μm/s.	Electric control	[15]
Walkers	2007	Cardiomyocytes	PDMS	Average step stroke: 77.6 mm; Speed: 100 μm/s.	No control	[78]
	2012	Cardiomyocytes	PEGDA	Per stroke: 354 µm; Speed: 236 μm/s.	No control	[79]
	2014	Skeletal muscles	PEGDA	Speed: 156 μm/s.	Electric control	[80]
	2016	Skeletal muscles	PEGDA	Speed: 310 μm/s.	Optical control	[81]
	2018	Skeletal muscles	PEGDA	Speed: 0.5 mm/s.	Electric control	[65]
	2019	Cardiomyocytes	CNT–GelMA	Speed: 20 μm/s.	Chemical control	[66]
	2021	Skeletal muscles	PEGDA	Speed: 5.9 mm/min.	Electric control	[82]
Grippers	2010	Skeletal muscles	PDMS	Manipulate objects sized: 200 µm; Displacement: ~8 µm.	Electric control	[67]
	2013	Skeletal muscles	PDMS	Displacement: ~5 µm.	Electric control	[68]
	2018	Skeletal muscles	Photo-reactive acrylate resin	Rotation angle: 90°.	Electric control	[43]
	2020	Skeletal muscles	Photo-reactive acrylate resin	Rotation angle: 90°.	Electric control	[52]
Pump-bots	2006	Cardiomyocytes	PDMS	Flow rate: 0.24 μL/min.	No control	[83]
	2007	Cardiomyocytes	PDMS; Cr/Au	Flow rate: 0.226 μL/min.	No control	[84]
	2007	Cardiomyocytes	PDMS	Flow rate: 0.047 μL/min.	No control	[85]
	2017	Cardiomyocytes	PDMS/FN	Flow rate: 6–8 μm/min.	No control	[71]
	2019	Cardiomyocytes	PDMS	Flow rate: 1.0 nL/min.	No control	[86]
	2019	Skeletal muscles	PDMS	Flow rate: 22.5 μL/min.	Electric control	[59]
	2022	Cardiomyocytes	Photoresist IP-S; PDMS	Flow rate: 0.3 μL/s.	No control	[16]
	2022	Skeletal muscles	PDMS	Flow rate: 13.62 μL/min.	No control	[69]

PDMS polydimethylsiloxane, PEG poly (ethylene glycol), CNT-GelMA carbon nanotubes-methacrylated gelatin, FN fibrinogen, PEGDA poly (ethylene glycol) diacrylate.

**Table 3 micromachines-14-01643-t003:** Comparison of the properties of extracellular materials.

Types of Extracellular Materials	Biocompatibility	Chemical Stability	Biotoxicity	Mechanical Property
Bioinert polymers	Excellent	Excellent	Low	Excellent
Bioactive hydrogels	Excellent	General	Low	General
Artificial hydrogels	General	Excellent	Low	Excellent
Tissue-harvested biomaterials	Excellent	General	Low	General

## Data Availability

Data sharing is not applicable to this article as no datasets were generated or analyzed during the current study.

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
