# Peer review of "Biohybrid Soft Robots Powered by Myocyte: Current Progress and Future Perspectives"

_micromachines, 2023, doi:10.3390/mi14081643_

Round 1

Reviewer 1 Report

Very comprehensive review! Different designs and problems of myocite based biohybrid robots are reviewed with numerous quuantative figures of their operation, with both advantages and drawbacks. 

The illustrations are good for such review, the literatyre is rather complete and contemporary.

Small revisions in English are needed.

Line 28 -"actuators" are repeated two times

L.78 "They have has remarkable" - "has" is excess word

L.84-85 - better "various advantages make an important direction for future research"

L. 127 element "-" is excess

L.260 add capital "T" to Tanaka

Author Response

Thank you for your letter and for the reviewer’s comments concerning our manuscript. Those comments are all valuable and very helpful for revising and improving our paper, as well as the important guiding significance to our researches. We have studied the comments carefully and have made correction which we hope meet with approval. Revised portion are marked in red in the paper. The main corrections in the paper and the responds to the reviewer’s comments are as follows:

Responds to the reviewer’s comments:

Reviewer #1:

Review report:

Very comprehensive review! Different designs and problems of myocite based biohybrid robots are reviewed with numerous quuantative figures of their operation, with both advantages and drawbacks.

The illustrations are good for such review, the literatyre is rather complete and contemporary.

Small revisions in English are needed.

  1. Response to comment:

Line 28 -"actuators" are repeated two times

Response: We thank the reviewer for the comment.

Based on the suggestions of the reviewers, we have corrected the errors here.

Lines:28

The component structures of robots are actuators, control systems, and sensing devices.

  1. Response to comment:

L.78 "They have has remarkable" - "has" is excess word

Response: We thank the reviewer for the comment.

Based on the suggestions of the reviewers, we have corrected the errors here.

Lines:78

They have remarkable controllability, output force, and power density, as well as self-assembly, self-healing ability, and better biocompatibility.

  1. Response to comment:

L.84-85 - better "various advantages make an important direction for future research"

Response: We thank the reviewer for the comment.

Based on the suggestions of the reviewers, we have made modifications here.

Lines: 84-85

The various advantages make an important direction for future research (Figure 1B,C).

  1. Response to comment:
  2. 127 element "-" is excess

Response: We thank the reviewer for the comment.

Based on the suggestions of the reviewers, we have corrected the errors here.

Lines:127

Microelectrode arrays (MEAs) are biosensors that can be used to monitor the electrophysiological activity of cardiomyocytes.

  1. Response to comment:

L.260 add capital "T" to Tanaka

Response: We thank the reviewer for the comment.

Based on the suggestions of the reviewers, we have corrected the errors here.

Lines:260

Tanaka et al. poured a suspension of neonatal rat cardiomyocytes into PDMS micromolds and cultured them for 7 days.

Finally, we express our sincere appreciation to the reviewers for their insightful viewpoints regarding this paper, which provided critical yet constructive comments and suggestions for us to improve the quality of our manuscript. We hope that our explanations and revisions have satisfactorily addressed the questions raised by the reviewers.

Reviewer 2 Report

Yuan et al. provide a comprehensive overview and insightful discussion of the current landscape and potential of myocyte-driven robots, including four key dimensions: myoblast types, extracellular materials, control methods, and motion profiles. The paper demonstrates a commendable structure and a coherent narrative flow. The reviewer recommends the paper for publication after addressing the following questions and comments:

1.     Figures should be better organized and presented. The current form of all figures appears to be derived from a simple copy-and-paste from existing literature.

2.     Inconsistent word usage is observed, such as the term "behavior" in Table 2 and "robot motion" in Figure 2, as well as the usage of "abiotic material" in the first paragraph of Chapter 3 and "Extracellular materials."

3.     A discrepancy arises where line 275 asserts that bioinert polymers exhibit poor biocompatibility, while Table 3 suggests that bioinert polymers possess excellent biocompatibility.

4.     The paper would benefit from more in-depth discussions in the context of future research directions.

5.     Enhancing the clarity of Figure 5 could be achieved by categorizing the chemical structures based on different types of extracellular materials, contributing to better understanding.

NA

Author Response

Thank you for your letter and for the reviewer’s comments concerning our manuscript. Those comments are all valuable and very helpful for revising and improving our paper, as well as the important guiding significance to our researches. We have studied the comments carefully and have made correction which we hope meet with approval. Revised portion are marked in red in the paper. The main corrections in the paper and the responds to the reviewer’s comments are as follows:

Responds to the reviewer’s comments:

Reviewer #2:

Review report:

Yuan et al. provide a comprehensive overview and insightful discussion of the current landscape and potential of myocyte-driven robots, including four key dimensions: myoblast types, extracellular materials, control methods, and motion profiles. The paper demonstrates a commendable structure and a coherent narrative flow. The reviewer recommends the paper for publication after addressing the following questions and comments:

  1. Response to comment:

Figures should be better organized and presented. The current form of all figures

appears to be derived from a simple copy-and-paste from existing literature.

Response: We thank the reviewer for the comment.

In this review, we introduce two types of myocytes as a source of power for myocyte-driven robots, summarizing the cellular force, size, and controllability of cardiac and skeletal myocytes. Second, we discussed the extracellular materials used for myocyte-driven robots. The properties and fabrication methods of extracellular materials determine not only the performance of the myocytes (differentiation, contractility, and survivability) but also the performance of myocyte-driven robots (speed, force, and manipulation). Third, we reviewed the control methods applied to myocyte-driven robots, including electrical stimulation, optical stimulation, and chemical stimulation, and the advantages and disadvantages of each of them. Then the functions and applications of myocyte-driven robots were summarized according to their different modes of locomotion, including swimmers, walkers, grippers, and pump-bots, and we described their performance and development process.

Figures are organized and presented according to the organization of the article. And each figure was selected from the representative article to better introduce the contents of section. For instance, Figure 3 introduce the properties of cardiomyocytes. We selected four sub-figures from the representative article to describe action potentials within rat cardiomyocytes monolayers, the contraction amplitude of cardiomyocytes, and the contractile force of cardiomyocytes.

  1. Response to comment:

Inconsistent word usage is observed, such as the term "behavior" in Table 2 and "robot motion" in Figure 2, as well as the usage of "abiotic material" in the first paragraph of Chapter 3 and "Extracellular materials."

Response: We thank the reviewer for the comment.

Based on the suggestions of the reviewers, we have made modifications here.

Motion types

Time

Myocytes

Extracellular materials

Performance parameters

control methods

Reference

In addition to muscle tissue, another important component of myocyte-driven robots is extracellular materials.

Extracellular materials provide structural support, growth environment, and attachment substrate for myocytes.

Since myocytes need to be attached or embedded in extracellular materials to build adaptive and biomimetic hybrid robots.

Extracellular materials must meet specific requirements, including excellent biocompatibility, desirable mechanical properties, and a tunable microstructure.

The properties of several extracellular materials differ, and we compare them in Table 3.

  1. Response to comment:

A discrepancy arises where line 275 asserts that bioinert polymers exhibit poor

biocompatibility, while Table 3 suggests that bioinert polymers possess excellent

biocompatibility.

Response: We thank the reviewer for the comment.

Based on the suggestions of the reviewers, we have corrected the errors here.

The advantages of bioinert polymers make them very suitable as extracellular materials in myocyte-driven robots, which can be safely and effectively integrated with biological components.

  1. Response to comment:

The paper would benefit from more in-depth discussions in the context of future research directions.

Response: We thank the reviewer for the comment.

In the summary and perspectives part of the paper, we expound the future research direction of muscle cell driven robot in detail.

  1. Response to comment:

Enhancing the clarity of Figure 5 could be achieved by categorizing the chemical structures based on different types of extracellular materials, contributing to better understanding.

Response: We thank the reviewer for the comment.

Based on the suggestions of the reviewers, we made a classification of Figure 5 against the order in which extracellular material was introduced in the papers.

Finally, we express our sincere appreciation to the reviewers for their insightful viewpoints regarding this paper, which provided critical yet constructive comments and suggestions for us to improve the quality of our manuscript. We hope that our explanations and revisions have satisfactorily addressed the questions raised by the reviewers.

Reviewer 3 Report

This review article offers an insightful exploration into the evolution and categorizations of microrobots, placing particular emphasis on traditional, flexible material-driven, and biomaterial-driven robots. There is a significant focus on myocyte-driven robots, renowned for leveraging muscle tissues as a power source and celebrated for their efficiency and biomimetic characteristics. The main content of the review offers an exhaustive overview of the design, operation, utilities, and challenges intrinsic to myocyte-driven robotics. Overall, this article equips readers with a clear perspective on the latest advancements in biohybrid soft robots propelled by myocytes, marking its potential contribution to the domain. However, I have a few questions and recommendations:

1.     For both Figure 1 and Figure 2, the authors should reference the particular sources for each figure, unless they are original creations.

2.     In the introduction, the merits of myocyte-powered soft robots are highlighted. To enhance the depth and comprehensiveness of this review, I recommend the authors also touch upon the drawbacks of myocyte-powered soft robots. This would provide readers with a holistic understanding and direction for future research.

3.     In Table 1, kindly indicate the reference number corresponding to each robot type.

4.     In the section "2.1 Cardiomyocytes", the authors note, “Microscopic observations... observed[93].” I'd urge the authors to clarify if "66.34 bpm per minute" is intended. As a point of note, "bpm" inherently denotes "beats per minute", rendering "per minute" potentially superfluous.

5.     Still, in “2.1 Cardiomyocytes”, the statement: “When the myofilaments are grouped together... skeletal muscles[70]” seems ambiguous. Could the authors elucidate what is implied by "myofilaments group together to form myofilaments"?

6.     In the segment “4. Contraction of muscle tissue and control methods”, four stimulation methodologies are discussed. To my understanding, there are alternative stimulation techniques capable of actuating robot power via myocyte, such as thermal stimulation. I'd appreciate an explanation for its omission.

7.     In the section “5. Various applications of myocyte-driven robots”, besides the highlighted four applications, I suggest the authors ponder over and introduce other potential applications. It seems to me that such robots could also serve roles as Carriers, Tweezers or Micromanipulators, Biosensors, Drug Delivery Systems, among others.

Author Response

Thank you for your letter and for the reviewer’s comments concerning our manuscript. Those comments are all valuable and very helpful for revising and improving our paper, as well as the important guiding significance to our researches. We have studied the comments carefully and have made correction which we hope meet with approval. Revised portion are marked in red in the paper. The main corrections in the paper and the responds to the reviewer’s comments are as follows:

Responds to the reviewer’s comments:

Reviewer #3:

Review report:

This review article offers an insightful exploration into the evolution and categorizations of microrobots, placing particular emphasis on traditional, flexible material-driven, and biomaterial-driven robots. There is a significant focus on myocyte-driven robots, renowned for leveraging muscle tissues as a power source and celebrated for their efficiency and biomimetic characteristics. The main content of the review offers an exhaustive overview of the design, operation, utilities, and challenges intrinsic to myocyte-driven robotics. Overall, this article equips readers with a clear perspective on the latest advancements in biohybrid soft robots propelled by myocytes, marking its potential contribution to the domain. However, I have a few questions and recommendations:

  1. Response to comment:

For both Figure 1 and Figure 2, the authors should reference the particular sources for each figure, unless they are original creations.

Response: We thank the reviewer for the comment.

Based on the suggestions of the reviewers, we add specific sources to Figures 1 and 2.

Figure 1. The development of myocyte-driven robots in recent years. (A) Number of journal publications and citations of research on biohybrid robots in recent years. (B) Key word cloud map on myocyte-driven robots. (C) Examples of typical achievements in the development of myocyte-driven robots research. Self-assembled microdevices driven by muscle. Reproduced from Reference[40] with a permission from Nature Materials. A tissue-engineered jellyfish with biomimetic propulsion. Reproduced from Reference[41] with a permission from Nature Biotechnology. Phototactic guidance of a tissue-engineered soft-robotic ray. Reproduced from Reference[42] with a permission from Science. Biohybrid robot powered by an antagonistic pair of skeletal muscle tissues. Reproduced from Reference[43] with a permission from Science Robotics. Neuromuscular actuation of biohybrid motile bots. Reproduced from Reference[44] with a permission from Proc Natl Acad Sci U S A. An autonomously swimming biohybrid fish designed with human cardiac biophysics. Reproduced from Reference[45] with a permission from Science.

Figure 2. Overview of myocyte-driven robots, including cardiac muscle and skeletal muscle structures, extracellular materials, robot control methods, and types of robot motion. The micro hand was fabricated by PDMS using soft lithography Reproduced from Reference[67] with a permission from 2010 3rd IEEE RAS & EMBS International Conference on Biomedical Robotics and Biomechatronics. PEDOT/MWCNT sheets Reproduced from Reference[72] with a permission from Scientific Reports. A model consisting of two different micropatterns of PEG and CNT-GelMA hydrogel layers Reproduced from Reference[73] with a permission from Adv Mater. PEGDA skeleton Reproduced from Reference[65] with a permission from Advanced Functional Materials. An autonomously swimming biohybrid fish designed with human cardiac biophysics. Reproduced from Reference[45] with a permission from Science. Bioinspired soft robotic caterpillar with cardiomyocyte drivers Reproduced from Reference[66] with a permission from Advanced Functional Materials. Biohybrid robot powered by an antagonistic pair of skeletal muscle tissues. Reproduced from Reference[43] with a permission from Science Robotics. A valveless pump robot powered by engineered skeletal muscles. Reproduced from Reference[59] with a permission from Proceedings of the National Academy of Sciences.

  1. Response to comment:

In the introduction, the merits of myocyte-powered soft robots are highlighted. To enhance the depth and comprehensiveness of this review, I recommend the authors also touch upon the drawbacks of myocyte-powered soft robots. This would provide readers with a holistic understanding and direction for future research.

Response: We thank the reviewer for the comment.

Based on the suggestions of the reviewers, we add the drawbacks of myocyte-driven robots in the introduction.

While myocyte-driven robots offer exciting possibilities for integrating biological and mechanical systems, they also present significant challenges in terms of control, maintenance, durability, and ethical considerations. The field is still in its early stages, and ongoing research is addressing these and other questions.

  1. Response to comment:

In Table 1, kindly indicate the reference number corresponding to each robot type.

Response: We thank the reviewer for the comment. According to reviewer’s suggestion, we have revised Table 1 and added the reference number.

Table 1. Advantages and disadvantages of the three types of robots

Robot types

Advantage

Disadvantage

Reference

Traditional rigid robots

High output power; High speed; High accuracy; Easy manipulation

Complex structure; less flexible; poor reliability; Low energy conversion rate

        [19-21]

Flexible material-driven robots

Light weight; High adaptability to target shapes; High flexibility

Low lifetime; Inefficient movement

        [22-27]

Biomaterial-driven robots

Excellent biocompatibility; High sensitivity; High stability; High energy conversion rate; Self-assembly and self-healing capability

Low lifetime; Ethical Issues; Cell survival environment issues; Simple function

        [36-39]

  1. Response to comment:

In the section "2.1 Cardiomyocytes", the authors note, “Microscopic observations... observed[93].” I'd urge the authors to clarify if "66.34 bpm per minute" is intended. As a point of note, "bpm" inherently denotes "beats per minute", rendering "per minute" potentially superfluous.

Response: We thank the reviewer for the comment.

Based on the suggestions of the reviewers, we have corrected the errors here.

Microscopic observations of the beating process of free stem-cell-derived cardiomyocytes allowed an assessment of the detailed deformation of individual cardiomyocytes, and a beating frequency of up to 66.34 bpm was observed.

  1. Response to comment:

Still, in “2.1 Cardiomyocytes”, the statement: “When the myofilaments are grouped together... skeletal muscles[70]” seems ambiguous. Could the authors elucidate what is implied by "myofilaments group together to form myofilaments"?

Response: We thank the reviewer for the comment.

Based on the suggestions of the reviewers, we have corrected the errors here.

When myofilaments are grouped together in a very ordered pattern, a sarcomere is formed, which is the basic contractile unit of skeletal muscle.

  1. Response to comment:

In the segment “4. Contraction of muscle tissue and control methods”, four stimulation methodologies are discussed. To my understanding, there are alternative stimulation techniques capable of actuating robot power via myocyte, such as thermal stimulation. I'd appreciate an explanation for its omission.

Response: We thank the reviewer for the comment.

In terms of contraction and control methods of muscle tissue, we mainly collate and introduce the current common stimulation mode of muscle cell-driven robots, other stimuli are less commonly used in the control of muscle cell-driven robots, so we do not introduce them in detail.

  1. Response to comment:

In the section “5. Various applications of myocyte-driven robots”, besides the highlighted four applications, I suggest the authors ponder over and introduce other potential applications. It seems to me that such robots could also serve roles as Carriers, Tweezers or Micromanipulators, Biosensors, Drug Delivery Systems, among others.

Response: We thank the reviewer for the comment.

Based on the suggestions of the reviewers, we supplemented this in the original paper.

Major applications of muscle cell-driven robots include muscle cell-driven swimmers, walkers, grippers, and pump robots. At the same time, it can also be used as a carrier, tweezers or micromanipulators, biosensors, drug delivery systems, etc.

Finally, we express our sincere appreciation to the reviewers for their insightful viewpoints regarding this paper, which provided critical yet constructive comments and suggestions for us to improve the quality of our manuscript. We hope that our explanations and revisions have satisfactorily addressed the questions raised by the reviewers.

Round 2

Reviewer 3 Report

All queries and recommendations have been adequately addressed. The paper's quality has seen substantial enhancement.